# OpenReview forum: "SafeVision: Efficient Image Guardrail with Robust Policy Adherence and Explainability"
_ICLR.cc/2025/Conference — Submitted to ICLR 2025_

### Official Review · Reviewer_CZEm · 2024-10-28

**Soundness:** 4
**Presentation:** 3
**Contribution:** 4
**Rating:** 6
**Confidence:** 3

**Summary:**

This work explores multimodal large models for detecting harmful content. Existing work suffers from low generalization and difficulty in handling new categories of hazards. Therefore, the authors built VISIONHARM-500K, a high-quality unsafe image benchmark comprising over 500k images to cover a wide array of risky categories. Based on this benchmark, the authors proposed SAFEVISION, which supports multiple modes and provides precise explanations. Experiments show the effectiveness and efficiency of the proposed method.

**Strengths:**

1. This paper constructs VISIONHARM-500K, which is conducive to the further development of image moderation.

2. The proposed SAFEVISION can better distinguish harmful content and has extremely high efficiency.

**Weaknesses:**

1. The categories defined by the dataset should be reflected in the main body.
2. How does SAFEVISION judge the content of new categories? Is it controlled only by prompts?
3. In Section 4.2, is the improvement to the tokenizer to add category nouns to the vocabulary library?
4. This work declares the contribution of the dataset, but there is no related open-source plan in the text. Will the code and dataset of this paper be released?

**Questions:**

See Weaknesses.

---

> ### Author Response · Authors · 2024-11-24
> **Response to Reviewer CZEm**
>
> Thank you very much for your valuable suggestions and thoughtful feedback. We appreciate your recognition of the superior performance of SafeVison and the contribution of the VisionHARM-500K dataset. Below, we address the specific weaknesses and questions point by point and hope these can help address your concerns.
>
> ***Q1:** The categories defined by the dataset should be reflected in the main body.*
>
> **A1:**  Thank you for this valuable feedback. We will add a detailed description of VISIONHARM-500K's categories in Section 3 in our revised submission, VISIONHARM-500K dataset includes 10 categories: Safe, Hate Humiliation Harassment, Violence Harm Cruelty, Sexual, Criminal Planning, Weapons Substance Abuse, Self Harm, Animal Cruelty, Disasters Emergencies, and Political.
>
> ***Q2**: How does SAFEVISION judge the content of new categories? Is it controlled only by prompts?*
>
> **A2**: As shown in our evaluation in Section 5.5.3, using only simple prompt changes will cause performance degradation. SAFEVISION's handling of new categories extends beyond simple prompt control. Our system employs 1) A **dynamic policy framework** that allows a flexible definition of new categories through structured guardrail policies, 2) A **text-based few-shot learning** approach that leverages our pretrained multimodal representations. The prompt template in Appendix A.5 is just one component of this comprehensive system. We will clarify this design in the revised paper.
>
> ***Q3**: In Section 4.2, is the improvement to the tokenizer to add category nouns to the vocabulary library?*
>
> **A3**: The tokenizer enhancement encompasses two key innovations: 1) **class tokens** for the predefined unsafe categories. 2) **structural tokens** for faster and more stable formative response generation. Such tokenizer redesign contributes significantly to both accuracy and inference speed improvements. We will expand this discussion in Section 4.2.
>
> ***Q4**: This work declares the contribution of the dataset, but there is no related open-source plan in the text. Will the code and dataset of this paper be released?*
>
> **A4**: Yes, we are committed to fostering reproducible research. Our release plan includes: 1) The complete VISIONHARM-500K dataset with detailed documentation, 2) SAFEVISION's model implementation and training code, 3) Evaluation scripts and benchmarking tools. We will make these resources available through a public GitHub and Huggingface repository as soon as the anonymity ends.

---

> ### Author Response · Authors · 2024-11-27
>
> Dear reviewer CZEm:
>
> We sincerely appreciate your thoughtful feedback and the time you have invested in reviewing our paper. Please let us know if you have any further suggestions or comments. If there are any additional questions or issues you'd like to discuss, we are fully committed to engaging further to enhance our paper.
>
> Best regards,
>
> Authors

---

### Official Review · Reviewer_frKA · 2024-10-30

**Soundness:** 2
**Presentation:** 1
**Contribution:** 2
**Rating:** 5
**Confidence:** 4

**Summary:**

The paper proposes SafeVision, a novel image guardrail system based on Vision-Language Models to detect and comprehend synthetic images. The system consists of two key modules: fast classification for general filtering scenarios and multimodal comprehension for policy-specified guardrails. The framework is evaluated with multiple safeguard scenarios and compared with diverse models. The proposed dataset is sufficiently scalable compared to existing datasets.

**Strengths:**

1. The research topic of safeguarding image generators is important and intriguing.
2. The proposed dataset is novel with good contributions.
3. The evaluation of various existing safeguarding methods is fair.

**Weaknesses:**

1. The proposed self-refinement training involves a testing procedure where the new version of the model is evaluated on the test set and misclassified instances are extracted & analyzed to curate new policies (Line 267 - Line 269). The paper should argue why and how such a procedure avoids **test set leakage**. If the test set information is encoded in the renewed policy, the procedure could lead to inflated performance on the test set because the model has already captured the exact test information during training. Also, the paper should compare how existing works handle misclassified instances in the test set.

2. The paper presentation needs significant improvement. For example, Appendix C.5 refers readers to a null table (Line 1407) and the prompt description on page 20 exceeds the page boundary (Line 1026 - Line 1079). The quotation marks in the prompt description are often monotonously right quotations (page 16, page 19, page 22). The reference list is also not well-curated. For example, the referenced website addresses often far exceed the page boundary and are obscured (Line 665, Line 778). Note that the Llama-Guard Team's paper is referenced as "Team (2024)" in the paper (Line 144, Line 215, ...), which reads strange. Also, there are many lines of unspecified space on page 18 (Line 922 - Line 958). **While a few minor errors in the paper presentation will not affect the rating, **too frequent observation** of them will harm the rating since they are not aligned with the proceeding guidelines and are not beneficial for future readers.**

**Questions:**

Please address my concerns stated in the weakness section. Although the novelty and presentation are limited, I still appreciate the contribution of this new dataset for image safeguarding. I give this submission an initial rating of borderline reject, and I look forward to the authors' response.

---

> ### Author Response · Authors · 2024-11-24
> **Response to Reviewer frKA**
>
> Thank you very much for your valuable suggestions and thoughtful feedback. We appreciate your recognition of the research topic of SafeVison and the contribution of VisionHARM-500K dataset. Below, we address the specific weaknesses and questions point by point and hope these can help address your concerns.
>
> ***Q1**: The proposed self-refinement training involves a testing procedure where the new version of the model is evaluated on the test set and misclassified instances are extracted & analyzed to curate new policies (Line 267 - Line 269). The paper should argue why and how such a procedure avoids test set leakage. If the test set information is encoded in the renewed policy, the procedure could lead to inflated performance on the test set because the model has already captured the exact test information during training. Also, the paper should compare how existing works handle misclassified instances in the test set.*
>
> **A1**:  We appreciate this important concern about potential test set leakage. To clarify our methodology:
>
> - We maintain three distinct datasets: **training, validation, and test sets**. The self-refinement process utilizes only the validation set, not the test set. The test set is completely isolated and is used solely for final evaluation.
>
> - For the policy refinement process, misclassified instances are identified exclusively using the validation set. Policy updates are made based on feedback from the validation set, ensuring that the test set remains untouched throughout the entire training process.
>
> Additionally, existing guardrail models do not handle misclassified instances in their training or validation processes. Such a refinement process is a unique contribution of SafeVision. The dynamic approach offers several advantages:
>
> - Better handling of edge cases through policy evolution.
> - Improved generalization to new categories.
>
> - More robust policy adaptation capabilities.
>
> We will clarify this approach and the difference with the existing method in the revised paper.
>
> ***Q2:** The paper presentation needs significant improvement. For example, Appendix C.5 refers readers to a null table (Line 1407) and the prompt description on page 20 exceeds the page boundary (Line 1026 - Line 1079). The quotation marks in the prompt description are often monotonously right quotations (page 16, page 19, page 22). The reference list is also not well-curated. For example, the referenced website addresses often far exceed the page boundary and are obscured (Line 665, Line 778). Note that the Llama-Guard Team's paper is referenced as "Team (2024)" in the paper (Line 144, Line 215, ...), which reads strange. Also, there are many lines of unspecified space on page 18 (Line 922 - Line 958). While a few minor errors in the paper presentation will not affect the rating, too frequent observation of them will harm the rating since they are not aligned with the proceeding guidelines and are not beneficial for future readers.*
>
> **A2:** Thank you for your thorough review of our paper's presentation. We have made comprehensive improvements:
>
> **Technical Corrections:**
>
> Fixed the missing table in Appendix C.5
>
> Adjusted prompt descriptions to fit within page boundaries
>
> Standardized quotation marks throughout the document
>
> Properly formatted website references with appropriate line breaks
>
> Updated the Llama-Guard Team citation to a more appropriate format
>
> Removed unnecessary spaces on page 18
>
> **Layout and Formatting:**
>
> Ensured all content fit within page boundaries
>
> Standardized formatting across all sections
>
> Improved reference formatting for better readability
>
> Fixed all spacing issues
>
> Please refer to our revised submission for these improvements.

---

> ### Author Response · Authors · 2024-11-27
>
> Dear reviewer frKA :
>
> We sincerely appreciate your thoughtful feedback and the time you have invested in reviewing our paper. Please let us know if you have any further suggestions or comments. If there are additional questions or issues you'd like to discuss, we are fully committed to engaging further to enhance our paper.
>
> Best regards,
>
> Authors

---

### Official Review · Reviewer_RhxE · 2024-10-31

**Soundness:** 2
**Presentation:** 3
**Contribution:** 3
**Rating:** 6
**Confidence:** 2

**Summary:**

This paper presents a dataset and model for detecting harmful visual content, with explanation and adaptation capability to new policies. The proposed method creates VisionHARM-500K dataset from LAION dataset, by using VLM filtering and image captioning. Built on the proposed VisionHARM-500K dataset, this paper also presents a model for detecting harmful content, which supports two modes. The first mode simply outputs classification results, and the second mode also generates a textual explanation with the harm score. To demonstrate the effectiveness, this paper compares the proposed method with many other models on binary classification, multi-classification, and new category harm classification. This paper presents stronger results in most settings than previous work. Overall, the paper is clearly written, but I have some concerns on the details, especially in the comparison with other methods. Therefore, I lean to reject this work slightly, even though this paper presents a lot. I could change my mind after rebuttal.

**Strengths:**

+ This paper presents a new dataset, VisionHARM-500K dataset, which can be used by the community to study vision safety problem. Researchers can also utilize the results presented in this paper as baseline, to move forward and achieve better results.

+ This paper presents better results than many previous work in terms of binary classification and multi-classification tasks. Ablation studies are also presented.

**Weaknesses:**

- This paper shows fewer novelties or contributions in model architecture for VLM learning and inference. In addition, this paper discusses very few about the network architecture. I prefer to learn more about model details about the proposed method. I cannot see how vision encoder and policy prompt encoder fuse each other.

- The experiments are not very convincing. For the comparing methods, how do those models train? Are those model trained on VisionHARM-500K dataset? It is unclear that the proposed method is significantly better than other methods, because of potential data distribution bias. From Figure 5, I am not convinced that the proposed model is significantly better than other models, as the proposed method trained on VisionHARM-500K and tested on VisionHARM-500K. In new category experiment, we can see GPT-4o is better than this method.

- In Table 1 & 2, citations are necessary for each comparing method.

**Questions:**

See weakness.

---

> ### Author Response · Authors · 2024-11-24
> **Response to Reviewer RhxE (Part1)**
>
> Thank you very much for your valuable suggestions and thoughtful feedback. We appreciate your recognition of the superior performance of SafeVison and the contribution of VisionHARM-500K dataset. Below, we address the specific weaknesses and questions point by point and hope these can help address your concerns.
>
> ***Q1**: This paper shows fewer novelties or contributions in model architecture for VLM learning and inference. In addition, this paper discusses very few about the network architecture. I prefer to learn more about model details about the proposed method. I cannot see how vision encoder and policy prompt encoder fuse each other.*
>
> **A1**:  We appreciate the reviewer's comments and would like to clarify that our paper's focus deliberately avoids architectural modifications for several reasons:
>
> - Architectural modifications would require training large-scale VLMs from scratch—an approach that is both computationally intensive and resource-prohibitive. In SafeVision, a unique contribution is our training approach that preserves the pre-trained capabilities of the VLM model while enhancing its guardrail abilities.
>
> - More importantly, our pipeline is designed to be model-agnostic and forward-compatible. While architectural changes often become model-specific and lack transferability, our methodology can be adapted to enhance future VLMs into better guardrail models.
>
> - Even without modifying the model architecture, we achieve state-of-the-art performance through innovations in data quality (diverse QA pairs), training methodology (novel self-refinement), and loss function design (carefully weighted objectives). This demonstrates that our approach brings significant novelty to existing guardrail models while maintaining architectural simplicity.
>
> Regarding the fusion between vision and policy prompt encoders, we leverage the proven methodology from our backbone model, InternVL2 [1], utilizing QLLaMA as a language middleware to align visual and linguistic features. This choice maintains consistency with established approaches while allowing us to fully leverage the well-trained capabilities of the base model.
>
> [1] Chen, Zhe, et al. "Internvl: Scaling up vision foundation models and aligning for generic visual-linguistic tasks." Proceedings of the IEEE/CVF Conference on Computer Vision and Pattern Recognition. 2024.
>
> ***Q2.1**: The experiments are not very convincing. For the comparing methods, how do those models train? Are those models trained on VisionHARM-500K dataset? It is unclear that the proposed method is significantly better than other methods, because of potential data distribution bias.*
>
> **A2.1**: Thank you for raising this valuable question! To clarify, we used the baseline models' pre-trained weights directly without fine-tuning them on the VisionHARM-500K dataset. We want to highlight that such a choice is because the VISIONHARM-500K dataset is a part of our proposed method and the key contributions of our work.
>
> We also acknowledge that our evaluation does not have conclusively proven the superiority of our training pipeline and dataset. To address this concern and provide a more comprehensive analysis, we conducted an ablation study as per your suggestion under three settings:
>
> - using the VisionHARM-500K dataset without our training pipeline
>
> - using our training pipeline with the dataset from Llavaguard
>
> - using both the VisionHARM-500K dataset and our training pipeline
>
> We use accuracy to evaluate the performance of each model.  The results are shown in the table below.
>
> |     Model      | Baseline | VISIONHARM-500K without training pipeline | Llavaguard train set + training pipeline | VISIONHARM-500K + training pipeline |
> | :------------: | :------: | :---------------------------------------: | :--------------------------------------: | :---------------------------------: |
> | Llavaguard-13b |  68.9%   |                   85.7%                   |                  74.4%                   |                93.0%                |
> |  Internvl2-2b  |  36.9%   |                   63.1%                   |                  73.4%                   |                91.8%                |
>
> The results show that even when using the Llavaguard train set instead of VISIONHARM-500K, the model still generates better results with our training pipeline. For instance, the performance of internvl2-2b improves **from 36.9% to 73.4%** when trained on the Llavaguard train set using our pipeline, surpassing its performance when trained on VISIONHARM-500K without the pipeline (63.1%). This suggests that the training pipeline contributes more to the performance than the dataset itself. However, the best performance is achieved when both VISIONHARM-500K and the training pipeline are used together.

---

> ### Author Response · Authors · 2024-11-24
> **Response to Reviewer RhxE (Part2)**
>
> ***Q2.2**: From Figure 5, I am not convinced that the proposed model is significantly better than other models, as the proposed method trained on VisionHARM-500K and tested on VisionHARM-500K. In the new category experiment, we can see  GPT-4o is better than this method.*
>
> **A2.2**:  Thank you for your question. We would like to clarify that the purpose of Table 5 is to demonstrate that the SafeVision training framework does not hurt the performance of the model in unseen categories. Other VLM-as-guardrail methods (LLAVAGuard, LLamaGuard) generate **less than 0.2 F1 scores** in unseen categories, while SafeVision maintains the performance of the original VLM backbone (InterVL-8B) and even achieves a better average performance. We will update the paper to clearly include our goal in our revision.
>
> Regarding the performance comparison with GPT-4O, we acknowledge that GPT-4O slightly outperforms SafeVision in some unseen categories(cult). However, it is important to note that SafeVision has **only 8B parameters,** and as evaluated in Table 2, the inference time overhead for SafeVision is **15 times** faster than GPT-4O. Moreover, SafeVision outperforms GPT-4O in trained categories. We believe we have successfully demonstrated the advantages of our method, which enables smaller models to achieve performance exceeding the existing state-of-the-art models on existing categories through low-cost fine-tuning, while maintaining the model's performance in untrained categories and greatly improving the model's efficiency.
>
> To further evaluate the model's performance in unseen categories, we conducted an extra evaluation using several public datasets containing novel categories: Bullying[1], Guns[2], Bloody[3], Fire[4], Alcohol[5], Cocaine[5], and Tobacco[5]. This resulted in a large test set containing 3,223 images. We use the F1 score to evaluate the performance of each model. The results are shown in the table below:
>
> | Model/Category | Safe      | Alcohol   | Bloody    | Bullying  | Cocaine   | Fire      | Guns      | Average   |
> | -------------- | --------- | --------- | --------- | --------- | --------- | --------- | --------- | --------- |
> | LLamaGuard3    | 0.258     | 0.086     | 0.685     | 0.000     | 0.000     | 0.099     | 0.000     | 0.349     |
> | LLaVAGuard     | 0.272     | 0.836     | 0.000     | 0.000     | 0.095     | 0.018     | 0.025     | 0.077     |
> | GPT-4o         | 0.680     | **0.932** | 0.942     | 0.453     | 0.773     | 0.671     | 0.997     | 0.908     |
> | InternVL2      | 0.649     | 0.721     | 0.810     | 0.377     | 0.780     | 0.743     | 0.994     | 0.892     |
> | SafeVision-8B  | **0.727** | 0.887     | **0.961** | **0.504** | **0.824** | **0.789** | **0.997** | **0.929** |
>
> SafeVision outperforms GPT-4O in all categories except for a slight lag in the Alcohol category. We hope our response clarifies our evaluation goal and demonstrates the performance comparison with GPT-4O, addressing your concerns.
>
> [1] [https://huggingface.co/datasets/Zoooora/BullyingAct](https://huggingface.co/datasets/Zoooora/BullyingAct)
>
> [2] [https://huggingface.co/datasets/JoseArmando07/gun-dataset](https://huggingface.co/datasets/JoseArmando07/gun-dataset)
>
> [3] [https://huggingface.co/datasets/NeuralShell/Gore-Blood-Dataset-v1.0](https://huggingface.co/datasets/NeuralShell/Gore-Blood-Dataset-v1.0)
>
> [4] [https://huggingface.co/datasets/EdBianchi/SmokeFire](https://huggingface.co/datasets/EdBianchi/SmokeFire)
>
> [5] [https://huggingface.co/datasets/luisf1xc/data_drugs_class](https://huggingface.co/datasets/luisf1xc/data_drugs_class)
>
> ***Q3**: In Table 1 & 2, citations are necessary for each comparing method.*
>
> **A3**: Thank you for pointing this out, we will add citations for all the baselines and datasets in Table 1 & 2. Please refer to our updated paper for these modifications.

---

> ### Author Response · Authors · 2024-11-27
>
> Dear Reviewer RhxE:
>
> We sincerely appreciate your thoughtful feedback and the time you have dedicated to reviewing our paper. Please let us know if you have any further suggestions or comments. If there are any additional questions or issues you would like to discuss, we are fully committed to engaging further to enhance our paper.
>
> Best regards,
>
> Authors

---

> > ### Comment · Reviewer_RhxE · 2024-12-02
> > **Response to authors - Reviewer RhxE**
> >
> > Thanks for the feedback and additional efforts in the rebuttal. After reading the feedback and revised version on the novelty explanation and extra experiments, I decide to upgrade my rating to "6: marginally above the acceptance threshold", even though I still don't believe the novelty of this work is its value. The authors need to clarify the novelty and model details in the experiments (maybe in appendix) for a strong publication, if this work is accepted. Good luck.

---

> > > ### Author Response · Authors · 2024-12-03
> > >
> > > Dear Reviewer RhxE,
> > >
> > > Thank you very much for your thoughtful feedback and for taking the time to review our work. We sincerely appreciate your recognition of our efforts in addressing your concerns. If our paper is accepted, we will certainly further clarify the novelty and model details in our camera-ready version.
> > >
> > > Best regards,
> > >
> > > Authors

---

### Official Review · Reviewer_Kfxg · 2024-11-14

**Soundness:** 3
**Presentation:** 2
**Contribution:** 2
**Rating:** 5
**Confidence:** 4

**Summary:**

This paper proposes an unsafe image classification model, in which it particularly provides several features: 1) the classification results can be coupled with a human-readable explanation (i.e. why such image is classified as unsafe/harmful), 2) zero-shot ability to support user-defined novel classes (i.e. the text description/definition of the novel unsafe class), and 3) the model has fast inference time with having the output in JSON format. Moreover, this paper also proposes a VISIONHARM-500K dataset, which is large-scale, diverse (cover wide range of unsafe categories), and richly-annotated (e.g. explanations and QA-pairs) to support various training objectives.

**Strengths:**

+ The proposed method is experimentally shown to have superior performance with respect to different baselines (including four VLM guardrails and nine classifier guardrails) across several datasets (three multi-label datasets and six binary label datasets), with having better trade-off between model performance and computational overhead.

**Weaknesses:**

- The InternVL2-8B itself is already having comparable performance with respect to GPT-4o and the proposed method SafeVision-8B (which takes InternVL2-8B as its backbone) for the novel classes (cf. Figure 5), where the additional training procedure or even the model designs in the proposed method seem to not offer better zero-shot transferability.
- The comparison might be not fair enough, as the proposed SafeVision is trained on the proposed VISIONHARM-500K dataset where its harmful categories are actually the super-set (or union) of all the other multi-label and binary-label datasets. Moreover, the self-refinement training scheme used in the proposed method actually can be treated as an ensemble framework of utilizing the consensus of several strong VLMs (i.e. Qwen-VL-Chat, InternVL2-26B, LLaVA-v1.6-34B, and the continuously-updated SafeVision model). In summary, the proposed method adopts larger training set, leverages the ensemble framework during training, and is built upon a stronger backbone (i.e. InternVL2), it is hence not surprising to have superior performance than the other baselines, leading to potential concern of unfair comparison (perhaps there should be baselines of training the open-source VLM guardrails on the proposed dataset?).
- From the ablation study, it looks like the proposed method is quite sensitive to the hyper-parameter tuning (e.g. critical token weight ratio and the weights of VLMs in the self-refinement training scheme, while the value and the varying schedule of these hyper-parameters are manually set) and the format of few-shot samples.
- The design for the decoder of SafeVision for having fast inference is actually not new (i.e. having a list of special token in the tokenizer to improve the inference efficiency).
- The overall organization seems to be problematic, as the details of dataset collection and proposed method are mostly provided in the supplementary, leading to the concern of self-containment for the main paper.

**Questions:**

The authors should carefully address the concerns as listed in the weaknesses.

---

> ### Author Response · Authors · 2024-11-24
> **Response to Reviewer Kfxg (Part 1)**
>
> Thank you very much for your valuable suggestions and thoughtful feedback. We appreciate your recognition of the superior performance and lower computational overhead of SafeVison. Below, we address the specific weaknesses and questions point by point and hope these can help address your concerns.
>
> ***Q1**: The InternVL2-8B itself is already having comparable performance with respect to GPT-4o and the proposed method SafeVision-8B (which takes InternVL2-8B as its backbone) for the novel classes (cf. Figure 5), where the additional training procedure or even the model designs in the proposed method seem to not offer better zero-shot transferability.*
>
> **A1**: Thank you for your insightful comment. We agree that InternVL2-8B, which serves as the backbone for SafeVision-8B, demonstrates comparable performance to GPT-4O and SafeVision-8B in the novel classes, as shown in Figure 5. However, we would like to emphasize that the primary goal of the SafeVision training framework is not to improve zero-shot transferability but to enhance the model's performance in trained categories while maintaining its performance in untrained categories and significantly improving its efficiency. We will update the paper to further clarify this and add related discussion.
>
> As demonstrated in Tables 1 and 2, SafeVision-8B outperforms an even larger model from the same family (InternVL2-26B) and other baselines in trained categories across multiple independent datasets at **15 times faster** than the baseline.
>
> To further evaluate the model's performance in unseen categories, we conducted an extra evaluation using several public datasets containing novel categories following the reviewer’s suggestions: Bullying[1], Guns[2], Bloody[3], Fire[4], Alcohol[5], Cocaine[5], and Tobacco[5]. This resulted in a large test set containing 3,223 images. We use the F1 score to evaluate the performance of each model. The results are shown in the table below:
>
> | Model/Category | Safe      | Alcohol   | Bloody    | Bullying  | Cocaine   | Fire      | Guns      | Average   |
> | -------------- | --------- | --------- | --------- | --------- | --------- | --------- | --------- | --------- |
> | LLamaGuard3    | 0.258     | 0.086     | 0.685     | 0.000     | 0.000     | 0.099     | 0.000     | 0.349     |
> | LLaVAGuard     | 0.272     | 0.836     | 0.000     | 0.000     | 0.095     | 0.018     | 0.025     | 0.077     |
> | GPT-4o         | 0.680     | **0.932** | 0.942     | 0.453     | 0.773     | 0.671     | 0.997     | 0.908     |
> | InternVL2      | 0.649     | 0.721     | 0.810     | 0.377     | 0.780     | 0.743     | 0.994     | 0.892     |
> | SafeVision-8B  | **0.727** | 0.887     | **0.961** | **0.504** | **0.824** | **0.789** | **0.997** | **0.929** |
>
> SafeVision outperforms GPT-4O in all categories except for a slight lag in the Alcohol category. We hope our response clarifies our evaluation goal and demonstrates the performance comparison with GPT-4O, addressing your concerns.
>
> [1][https://huggingface.co/datasets/Zoooora/BullyingAct](https://huggingface.co/datasets/Zoooora/BullyingAct)
>
> [2][https://huggingface.co/datasets/JoseArmando07/gun-dataset](https://huggingface.co/datasets/JoseArmando07/gun-dataset)
>
> [3] [https://huggingface.co/datasets/NeuralShell/Gore-Blood-Dataset-v1.0](https://huggingface.co/datasets/NeuralShell/Gore-Blood-Dataset-v1.0)
>
> [4] [https://huggingface.co/datasets/EdBianchi/SmokeFire](https://huggingface.co/datasets/EdBianchi/SmokeFire)
>
> [5] [https://huggingface.co/datasets/luisf1xc/data_drugs_class](https://huggingface.co/datasets/luisf1xc/data_drugs_class)

---

> ### Author Response · Authors · 2024-11-24
> **Response to Reviewer Kfxg (Part 2)**
>
> ***Q2**: The comparison might be not fair enough...... Moreover, the self-refinement training scheme used in the proposed method actually can be treated as an ensemble framework......leading to potential concern of unfair comparison.*
>
> **A2**: Thank you for the valuable suggestions. We would like to address your concerns point by point:
>
> ***Q2.1**: The comparison might be not fair enough, as the proposed SafeVision is trained on the proposed VISIONHARM-500K dataset where its harmful categories are actually the super-set (or union) of all the other multi-label and binary-label datasets…. perhaps there should be baselines of training the open-source VLM guardrails on the proposed dataset?*
>
> **A2.1**: Thank you for your question! During the evaluation, we also evaluated SafeVision on the test splits from the datasets used to train the other models, as shown in Tables 1 & 2. The results demonstrate that our model consistently outperforms the other models even on their own test sets. For example, LlavaGuard achieves **0.688 accuracy** on the LlavaGuard dataset, while SafeVision attains **0.808 accuracy**. Although our dataset includes their categories, we didn't collect data from identical sources, so their train-test distributions should be more similar to each other than to our method. This just proves that our method and dataset have better generalizability.
>
> We acknowledge the importance of an ablation study to demonstrate the effectiveness of our training pipeline. We used the same backbone model fine-tuned under three different settings: (1) using the VISIONHARM-500K dataset without our training pipeline, (2) using our training pipeline with the LlavaGuard dataset, and (3) using both the VISIONHARM-500K dataset and our training pipeline. We use accuracy to evaluate the performance of each model. The results are shown in the table below:
>
> |     Model      | Baseline | VISIONHARM-500K without training pipeline | Llavaguard train set + training pipeline | VISIONHARM-500K + training pipeline |
> | :------------: | :------: | :---------------------------------------: | :--------------------------------------: | :---------------------------------: |
> | Llavaguard-13b |  68.9%   |                   85.7%                   |                  74.4%                   |                93.0%                |
> |  Internvl2-2b  |  36.9%   |                   63.1%                   |                  73.4%                   |                91.8%                |
>
> The results show that even when using the Llavaguard train set instead of VISIONHARM-500K, the model still generates better results with our training pipeline. For instance, the performance of internvl2-2b improves **from 36.9% to 73.4%** when trained on the Llavaguard train set using our pipeline, surpassing its performance when trained on VISIONHARM-500K without the pipeline (63.1%). This suggests that the training pipeline contributes more to the performance than the dataset itself. However, the best performance is achieved when both VISIONHARM-500K and the training pipeline are used together.
>
> ***Q2.2**: Moreover, the self-refinement training scheme used in the proposed method actually can be treated as an ensemble framework of utilizing the consensus of several strong VLMs (i.e. Qwen-VL-Chat, InternVL2-26B, LLaVA-v1.6-34B, and the continuously-updated SafeVision model)....leverages the ensemble framework during training, and is built upon a stronger backbone (i.e. InternVL2), it is hence not surprising to have superior performance than the other baselines*
>
> **A2.2**: We greatly appreciate the valuable comments! To explain, while the models used in our self-refinement training scheme may be stronger on some general task benchmarks, Tables 1 and 2 show that their individual performance on the image guardrail task is not stronger than the final model we trained. This highlights that our method has achieved better performance by using only weaker models, which can be considered a significant contribution. Furthermore, we only use InternVL2-2B and InternVL2-8B as the backbone, which are smaller models, yet they still outperform InternVL2-26B after training.
>
> In addition to performance, our proposed method also increases the efficiency of the model for the guardrail task, making it an important contribution that enables low-cost deployment in real-world industry applications.
>
> Moreover, since our proposed method consists of a dataset and training pipeline, it can be adopted by almost all VLM models. This means that when stronger models become available in the future, we can replace the current backbone with a better model, making our method scalable.

---

> ### Author Response · Authors · 2024-11-24
> **Response to Reviewer Kfxg(Part 3)**
>
> ***Q3**: From the ablation study, it looks like the proposed method is quite sensitive to the hyper-parameter tuning (e.g. critical token weight ratio and the weights of VLMs in the self-refinement training scheme, while the value and the varying schedule of these hyper-parameters are manually set) and the format of few-shot samples.*
>
> **A3**: Regarding the critical token weight ratio, our initial plot had a limited y-axis range of only 3 percent, and we only included SafeVision without any other baselines. In the updated figure, we have added GPT-4O and InternVL2 as baselines and tested both SafeVision-2B and SafeVision-8B. We also adjusted the y-axis scale to be more reasonable. Please refer to Section 5.5 Figure 6 for the updated figure. We can observe that while weighted loss does enhance model performance, the accuracy changes within a small range of less than 3 percent. This indicates that the model performance is actually quite stable.
>
> For the weights of different VLMs in the self-refinement training scheme, the weight for our model is set as $w \cdot \sqrt{\text{epoch}}$, while the other three VLMs share the same weight, calculated as $\frac{1 - w \times \sqrt{\text{epoch}}}{3}$. Therefore, the only parameter we need to adjust is $w$. Moreover , since $w$ can’t be too large initially or too small after several epochs, its range is actually limited. We have added a new experiment to demonstrate the influence of $w$, as shown in the table below. We set $w$ to 0.05,0.1,0.15 and 0.2 in the beginning and applied self-refinement training to a subset of the training data over multiple epochs and calculated the percent of remaining data after each epoch. From the results, it is evident that the data removed in each epoch is stable and not significantly affected by the choice of $w$.
>
> | Epoch | $w$ = 0.05 | $w$ = 0.10 | $w$ = 0.15 | $w$ = 0.20 |
> | ----- | ---------- | ---------- | ---------- | ---------- |
> | 0     | 100%       | 100%       | 100%       | 100%       |
> | 1     | 97.5%      | 98.5%      | 98.4%      | 98.3%      |
> | 2     | 96.5%      | 96.4%      | 96.2%      | 95.2%      |
> | 3     | 96.0%      | 95.2%      | 95.0%      | 93.9%      |
> | 4     | 94.7%      | 94.0%      | 94.1%      | 93.8%      |
>
> Regarding the format of few-shot samples, we found that using a detailed JSON format yields the best performance for SafeVision, so we continue to use this format in our evaluation. This is due to the inherent nature of the model and can be addressed through further model training, which we plan to explore in future work.
>
> ***Q4**: The design for the decoder of SafeVision for having fast inference is actually not new (i.e. having a list of special token in the tokenizer to improve the inference efficiency).*
>
> **A4**: Thank you for your comments. While we acknowledge that using special tokens to improve inference efficiency is not novel in general, we want to clarify that this is not our primary contribution. Importantly, our work is the first to apply this approach specifically within the image guardrail task, where it has led to substantial improvement in both performance and inference speed. This unique adaptation effectively addresses the task’s specialized requirements and complements our main contributions.
>
> ***Q5**: The overall organization seems to be problematic, as the details of dataset collection and proposed method are mostly provided in the supplementary, leading to the concern of self-containment for the main paper.*
>
> **A5**: Thank you for your valuable feedback. We acknowledge the concern regarding self-containment and agree that the dataset collection and proposed method sections could be better integrated into the main paper. We will revise these sections to ensure they are clearly presented within the main text. Please refer to our updated paper for these modifications.

---

> ### Author Response · Authors · 2024-11-27
>
> Dear Reviewer Kfxg:
>
> We sincerely appreciate your thoughtful feedback and the time you have invested in reviewing our paper. Please let us know if you have further suggestions or comments. If there are any additional questions or issues you'd like to discuss, we are fully committed to engaging further to enhance our paper.
>
> Best regards,
>
> Authors

---

### Author Response · Authors · 2024-11-24
**General Response**

We thank the reviewers for their insightful comments and constructive suggestions to improve our paper. We are pleased that the reviewers appreciated the importance of our research topic, recognized the novelty and quality of our VISIONHARM-500K dataset and guardrail model, and found our evaluation comparing SafeVision to existing safeguarding methods comprehensive and clear.

The reviewers raised some important points that we will address in detail in our response and paper revision:

1. We have designed additional experiments on new categories in our response. The results further demonstrate the effectiveness of our training pipeline in maintaining SafeVision’s performance in untrained categories while significantly improving its efficiency (**Reviewer Kfxg, RhxE**).
2. We have clarified the novelty of our training methodology and provided more comprehensive experiments and analysis in our response. The results highlight the effectiveness and superiority of our proposed pipeline and dataset (**Reviewer Kfxg, RhxE**).
3. We have included more detailed ablation studies on the hyper-parameters used in our training pipeline in our response and Section 5.5. The results show the robustness of SafeVision against variations in hyper-parameter choices (**Reviewer Kfxg**).
4. To address potential concerns about test set leakage, we have provided more details about our training pipeline, where the test set remains completely isolated and is used solely for final evaluation and misclassified instances will be identified exclusively using a separate validation set (**Reviewer frKA**).
5. We have compared our approach to handling misclassified instances with prior work. In our approach, we leverage misclassified instances to update the guardrail policy, addressing edge cases through policy evolution and improving the model’s generalization to new categories (**Reviewer frKA**).
6. We have clarified the novelty of our model architecture and analyzed the reason for deliberately avoiding architectural modifications (**Reviewer RhxE**).
7. We have enhanced the paper's presentation, formatting, and references based on the detailed feedback (**Reviewer Kfxg, frKA,CZEm**).
8. We have provided more details on the dataset categories, how SafeVision handles new categories beyond prompts, specifics of the tokenizer improvements, and our plans for open-sourcing the code and data (**Reviewer Kfxg, CZEm**).

We sincerely appreciate the reviewers' feedback and believe that these revisions have strengthened our paper. We hope our responses and the updated manuscript address all concerns and the reviewers will appreciate our work towards building safer AI systems.

---

### Meta-Review · Area_Chair_ZSUX · 2024-12-22

**Metareview:**

The paper proposes a new framework for guardrailing computer vision systems for improved safety. The proposed system effectively triages predictions for decision-making, such that for less certain predictions, a more human-like/chain-of-thought response must accompany the prediction. The response by the reviewers is mixed and mostly on the borderline negative side. During the post-rebuttal discussion phase, the reviewers reaffirmed that the evaluation is not rigorous enough, and that it is peculiar to have as many target improvement in the known classes but novel ones, since this is what you want from such a system. Thus, the paper does not pass the bar.

**Additional Comments On Reviewer Discussion:**

The design leading to faster inference for the decoder of SafeVision is not new, as admitted by the authors. Together with the fact that InternVL2 backbone itself already strikes a better balance between model size and performance, the claimed contribution of the proposed method in terms of both efficiency and accuracy is hence weak.

Concerns regarding using ensemble and larger dataset to achieve better performance (which is good and reasonable but can not be well considered as a significant contribution) are well resolved by the rebuttal, although the effort by the authors effort to provide additional experimental results is appreciated.

---

### Decision · Program_Chairs · 2025-01-22

Reject